# Autonomic Nerve Involvement in Post-Acute Sequelae of SARS-CoV-2 Syndrome (PASC)

**DOI:** 10.3390/jcm12010073

**Published:** 2022-12-22

**Authors:** Tae Hwan Chung, Antoine Azar

**Affiliations:** 1Department of Physical Medicine and Rehabilitation, Johns Hopkins University, Baltimore, MD 21224, USA; 2Department of Neurology, Johns Hopkins University, Baltimore, MD 21224, USA; 3Department of Medicine, Division of Allergy and Immunology, Johns Hopkins University, Baltimore, MD 21224, USA

**Keywords:** post-acute sequelae of SARS-CoV-2 syndrome (PASC), long COVID, post-COVID syndrome, postural orthostatic tachycardia syndrome (POTS), dysautonomia, autonomic nerve

## Abstract

The novel SARS-CoV-2 virus and resulting COVID-19 global pandemic emerged in 2019 and continues into 2022. While mortality from COVID-19 is slowly declining, a subset of patients have developed chronic, debilitating symptoms following complete recovery from acute infection with COVID-19. Termed as post-acute sequelae of SARS-CoV-2 syndrome (PASC), the underlying pathophysiology of PASC is still not well understood. Given the similarity between the clinical phenotypes of PASC and postural orthostatic tachycardia syndrome (POTS), it has been postulated that dysautonomia may play a role in the pathophysiology of PASC. However, there have been only a few studies that have examined autonomic function in PASC. In this retrospective study, we performed an analysis of autonomic nerve function testing in PASC patients and compared the results with those of POTS patients and healthy controls. Our results suggest that a significant number of PASC patients have abnormal autonomic function tests, and their clinical features are indistinguishable from POTS.

## 1. Introduction

The novel SARS-CoV-2 virus and resulting COVID-19 emerged in 2019, and the global COVID-19 pandemic continues into 2022. Since the vaccination against COVID-19 became available, the overall mortality rate has been declining [1,2,3,4]. However, a subset of patients have developed chronic, debilitating symptoms following complete recovery from acute infection with COVID-19. Multiple terms have been used to describe this constellation of symptoms, including long COVID, long-haul COVID, and post-acute sequelae of SARS-CoV-2 syndrome (PASC) [5]. Typical presentations include chronic fatigue, various gastrointestinal (GI) symptoms, brain fog, orthostatic tachycardia, and exertional dyspnea, among others. Symptoms can be severe and debilitating, and most patients experience disability or modified independence [6]. Despite the significance of these symptoms, the underlying pathophysiology of PASC remains largely unknown, and treatments are primarily focused on symptom management.

While PASC is likely a clinical syndrome with heterogeneous etiologies, including mast cell activation syndrome, viral persistence, and permanent tissue damage, increasing evidence suggests that autonomic dysfunction underlies the symptoms of PASC. Since the beginning of the pandemic, there have been several published case reports of postural orthostatic tachycardia syndrome (POTS) among patients with long-lasting symptoms after complete recovery from COVID-19 infection [7,8,9,10,11]. POTS is a clinical syndrome that is thought to be caused by an impaired vasomotor response secondary to autonomic dysfunction [12]. In a subset of POTS patients, there is evidence of sympathetic vasomotor denervation, which results in failure to increase systemic vascular resistance during orthostatic challenge or exercise. Compensatory central nervous system activation of the sympathetic nervous system leads to a hyperadrenergic reaction that presents as tachycardia, insomnia, GI dysfunction, and anxiety [12,13]. These are also the symptoms that are commonly observed in patients with PASC.

Evaluating autonomic nerve function in the clinical setting is challenging because the autonomic nervous system mainly consists of small fiber axons with slow conduction velocity and small amplitudes. There are very few clinical tools that can evaluate autonomic function, and only a few autonomic testing laboratories exist in the United States. Recently, Novak et al. demonstrated that nine patients with PASC referred to an autonomic testing center had evidence of cerebrovascular dysregulation and related dysautonomia [14]. The study also showed reduced small fiber and sudomotor density in cutaneous nerve biopsies from all the PASC patients that they recruited. However, in the study, none of the PASC patients had POTS, which is somewhat inconsistent with previous reports. Herein, we conducted a retrospective analysis of autonomic function tests of all PASC patients who were referred to the Johns Hopkins Autonomic Laboratory.

## 2. Materials and Methods

### 2.1. Study Design

This is a retrospective study of consecutive patients who were referred to the Johns Hopkins Autonomic Laboratory from August 2021 to August 2022. Patients were primarily recruited from the Johns Hopkins post-acute COVID-19 Clinic and Johns Hopkins POTS Clinic. PASC was diagnosed based on the CDC criteria where patients develop ongoing symptoms beyond 4–12 weeks after initial COVID-19 infection [15]. The severity of acute COVID-19 infection was defined as follows: mild, if no hospitalization was required; moderate, if hospitalization was required without an ICU stay; severe, if an ICU stay was required. Any PASC patients who showed orthostatic tachycardia or orthostatic hypotension were referred to the Johns Hopkins Autonomic Laboratory. Patients with PASC were included in the study if they met the following criteria: (1) aged 18 years old and above; (2) documented diagnosis of COVID-19 infection based on a positive reverse transcriptase–polymerase chain reaction (PCR) test or an antigen test for SARS-CoV-2; (3) completion of comprehensive autonomic testing at the Johns Hopkins Autonomic Laboratory; and (4) availability of electronic medical records at Johns Hopkins. Patients who were confirmed to have POTS during the same time period were included in the analyses if they met the following criteria: (1) aged 18 years old and above; (2) completion of autonomic testing at the Johns Hopkins Autonomic Laboratory; and (3) confirmed diagnosis of POTS at Johns Hopkins POTS Clinic. POTS was defined as having more than a 30-beats-per-minute (bpm) increase in heart rate that sustained at least 30% of the entire duration of tilt for more than 10 min if patients were older than 19 years old. For patients aged between 18 and 19, they had to have a 40 bpm increase in heart rate that sustained at least 30% of the entire duration of tilt for more than 10 min. In addition, the following conditions were required to confirm the diagnosis of POTS: (1) the symptoms were present for more than 3 months at least; (2) their typical symptoms were reproduced during the tilt table testing; and (3 other diagnoses with mimicking symptoms were ruled out. Healthy control subjects were recruited from the Johns Hopkins Autonomic Laboratory. They were referred to the Johns Hopkins Autonomic Laboratory for various reasons and ruled out for any other neurological diseases by referring physicians.

### 2.2. Tilt Table Testing

We utilized the full autonomic assessment laboratory equipment from WR Medical Co. (Maplewood, MN, USA) for heart rate variability during deep breathing, Valsalva maneuver, and tilt table test. A CNAP Monitor from CNsystem (Graz, Austria) was used for the continuous monitoring of blood pressure and heart rate from a CNAP finger sensor. Blood pressure and heart rate obtained from the finger sensor were calibrated to cuff measurements from the upper arm of the same arm. In the contralateral arm, an additional blood pressure cuff was placed 2–3 cm above the antecubital fossa and measured every 2 min for comparison. Heart rate was also monitored using 3-lead ECG electrodes. A chest expansion bellow was applied at the level of the xyphoid process. Tilt table testing was performed at 70° for at least 10 min after resting for more than 5 min. All the procedures were performed in the morning, and the patients were instructed to skip breakfast before testing. The procedure room was equipped with white noise and an adjustable light system to improve patient comfort and minimize anxiety.

### 2.3. Heart Rate Variability during Deep Breathing (HRVDB)

Patients were instructed to slowly inhale over 5 s and exhale over 5 s for 8 cycles. To assist with deep breathing, patients were instructed to follow arrows on monitors of the WR Medical equipment for inspiration and expiration. At least 2 cycles of 8 breaths were obtained for analysis. Average best heart rate differences between inspiration and expiration were calculated.

### 2.4. Valsalva Maneuver

Patients were provided with a mouthpiece connected to WR Medical equipment for the Valsalva maneuver. Patients were instructed to forcefully exhale to maintain pressure at 40 mmHg for 15 s. The pressure was monitored in real time on a monitor, and patients repeated the procedure at least 2–3 times. Mean blood pressure during Valsalva maneuver was used to evaluate for 4 phases. Phase I is a transient elevation in blood pressure due to increased intrathoracic pressure; early phase II is the decline in blood pressure due to reduced cardiac output; late phase II is the rise in blood pressure as a result of sympathetic compensation; phase III is a transient fall in blood pressure when the Valsalva maneuver is stopped; phase IV is an overshoot of blood pressure due to the persistent vasoconstriction that started in phase II. The Valsalva ratio was derived from the maximum heart rate generated by the Valsalva maneuver divided by the lowest heart rate occurring within 30 s of the peak heart rate.

### 2.5. Cutaneous Nerve Biopsy

Punch skin biopsies were performed using a 3 mm diameter circular biopsy instrument at distal, intermediate, and proximal leg sites. Biopsies were fixed in Zamboni fixative, cryoprotected, and sectioned using the free-floating technique. Four 50 µm sections at regular intervals throughout the sample with a random start point were selected and stained. Standard immunohistochemical staining was performed with rabbit anti-PGP9.5 (1:10,000, polyclonal; BioRad, Kidlington, UK). Intraepidermal nerve fiber density (IENFD, fibers/mm) in skin biopsies was determined using established counting rules; the count includes the number of nerve fibers crossing the epidermal basement membrane and isolated nerve fragments in the epidermis that do not cross the basement membrane [16]. The sweat gland nerve fiber density (SGNFD; length in m/mm^3^) was also determined. Both IENFD and SGNFD below the 5th percentile of age/sex-matched normative values were defined as being abnormal.

### 2.6. Antibody Testing

Blood samples were sent to Mayo Clinic Laboratory for a paraneoplastic panel, which included ganglionic acetylcholine receptor antibody, amphiphysin antibody, anti-glial nuclear antibody type 1–3, CRMP-5-IgG, neuronal (V-G) K+ Channel antibody, P/Q-type calcium channel antibody, and Purkinje cell cytoplasmic antibody type 1–3. Thyroid peroxidase antibody was also checked at the Johns Hopkins Laboratory when available.

### 2.7. Statistical Analysis

Student’s *t*-tests were performed to compare variables between POTS, PASC, and the control groups. Box and whisker plots were used to display the distribution of values of heart rate variability during deep breathing. Repeated-measures ANOVA was used to evaluate the effect of time during the tilt test. Graphpad Prism and Microsoft Excel software were used for statistical analysis.

## 3. Results

### 3.1. Demographics of Post-Acute Sequelae of SARS-CoV-2 Syndrome (PASC) and Postural Orthostatic Tachycardia Syndrome (POTS)

From August 2021 to August 2022, a total of 13 patients with PASC who presented with autonomic symptoms and 6 patients with POTS were referred for autonomic testing. In both POTS and PASC groups, patients were predominantly female and white. In PASC patients, 11/13 (84.62%) were female and 10/13 (76.92%) were white. In POTS patients, 5/6 (83.33%) were female and 4/6 (66.67%) were white. The average age of PASC patients was significantly higher than that of POTS (*t*-test, *p* = 0.01). Other demographic features, such as ratio of sex, ethnicity, and BMI between PASC and POTS, were not statistically different. The demographic features are summarized in Table 1.

### 3.2. Clinical Symptoms

In PASC patients, the symptoms of COVID-19 infection started with mild-to-moderate upper respiratory symptoms. A total of 2/13 (15.38%) patients had respiratory symptoms severe enough to require a short hospital stay (<5 days), but none required an ICU stay. All the other PASC patients had mild fever and upper respiratory symptoms that subsided within a few weeks. All PASC patients developed severe, chronic fatigue and orthostatic intolerance as their initial respiratory symptoms subsided after a few weeks to a few months following the initial symptom onset. The severity of symptoms waxed and waned over time. None of the PASC patients had similar symptoms of fatigue or orthostatic intolerance prior to COVID-19 infection. A total of 8/13 (61.54%) PASC patients complained of various gastrointestinal symptoms, including chronic nausea, occasional vomiting, indigestion, epigastric pain, acid reflux, constipation, and occasional diarrhea. A total of 11/13 (84.62%) PASC patients and 6/6 (100%) POTS patients had post-exertional malaise. In POTS patients, 2/6 (33.33%) patients reported precedent viral infection before the onset of chronic fatigue and orthostatic intolerance. All POTS patients developed chronic fatigue, orthostatic intolerance, and brain fog that waxed and waned over many years (median 87 months). A total of 3/6 (50%) POTS patients complained of various gastrointestinal symptoms that were similar to those of PASC patients. It is noteworthy that all POTS patients experienced some gastrointestinal symptoms at some point during their illness, although only 50% of them were complaining of gastrointestinal symptoms at the time of evaluation. All POTS patients reported post-exertional malaise. The clinical symptoms are summarized in Table 2. Regarding the presence of known autoantibodies, two out of seven patients with PASC who received antibody testing had known autoantibody markers (anti-TPO antibody and anti-ganglionic acetylcholine receptor (gAChR) antibody). All 6/6 POTS patients received antibody testing, and only 1/6 (16.67%) had a low-titer anti-gAChR antibody.

### 3.3. Tilt Table Test Response

Of the 13 patients with PASC, 2 patients could not complete the tilt table testing because 1 patient had a high body mass index (BMI) that exceeded the table manufacturer’s recommendation, and the other patient refused the tilt table test part of the autonomic testing due to anxiety. All 11 PASC patients who underwent the tilt table test showed significant orthostatic intolerance during the 70° tilt, which improved when they were placed back in the supine position (Figure 1). The average resting heart rate of PASC patients was 76.8 bpm, compared to 70.2 bpm for the controls. A total of 7/11 (63.64%) patients with PASC met criteria for POTS: they had more than a 30 bpm increase in heart rate that sustained for at least 30% of the entire tilt table duration. The remainder of the patients with PASC also had an increase in heart rate which did not reach 30 bpm or was not sustained. The heart rate changes during tilt were not statistically significant between PASC and POTS (*p* = 0.97) or between PASC and control (*p* = 0.07) in the repeated-measures ANOVA analyses. As expected, the heart rate changes during tilt showed significantly elevated heart rates in POTS compared to control (*p* = 0.03). The average mean blood pressure of the PASC patients was 87.6 mmHg, whereas those of the controls and POTS patients were 92.8 mmHg and 89.6 mmHg, respectively. During the tilt, there were no significant changes in blood pressure in all PASC, POTS, and control groups.

### 3.4. Heart Rate Variability during Deep Breathing

The average heart rate difference between inspiration and expiration in PASC patients was 15.1 bpm. A total of 11/13 (84.62%) patients with PASC had normal heart rate variability above the fifth percentile of age-matched normative values (Figure 2). However, 2/13 (15.38%) patients with PASC had a heart rate difference between inspiration and expiration lower than the fifth percentile of normative values, suggesting impaired cardiovagal reflex. All POTS patients showed normal heart rate differences between inspiration and expiration. The mean heart rate differences between inspiration and expiration were not significantly different between the POTS, PASC, and control groups.

### 3.5. Valsalva Ratio

A total of 11/13 (84.62%) patients with PASC showed normal Valsalva responses; the mean blood pressure showed normal phase I-IV morphologies, and heart rate responses (Valsalva ratio) were normal. However, 2/13 (15.38%) patients with PASC showed normal phase I-IV morphologies but abnormal Valsalva ratios, suggesting impaired vagal baroreflex response. Those two patients had abnormal HRVDB as described above, consistent with the abnormal Valsalva response. Valsalva responses of all POTS and control patients showed normal blood pressure and heart rate responses.

### 3.6. Cutaneous Nerve Biopsy

Among 13 PASC patients, 9 patients underwent cutaneous nerve biopsies that were processed at the Johns Hopkins Cutaneous Nerve Laboratory. A total of 5/9 (55.56%) patients showed reduced intraepidermal small fiber density lower than the fifth percentile of age/sex-matched normative values in at least more than one site. In total, 1 (11.11%) patient out of 9 had both reduced intraepidermal small fiber density and sweat gland innervation that were lower than the fifth percentile of age/sex-matched normative values. All other patients with PASC showed normal sweat gland innervation. In POTS patients, 5/6 cutaneous nerve biopsy reports were available for analysis. A total of 3/5 (60%) patients with POTS showed reduced intraepidermal small fiber density lower than the fifth percentile of age/sex-matched normative values in at least one site. Sweat gland innervation was normal in all POTS patients.

## 4. Discussion

In this retrospective study, clinical features and autonomic nerve function test results were indistinguishable between PASC and POTS patients. In both syndromes, the results of the autonomic function tests suggested vasomotor dysfunction, which explains many symptoms such as orthostatic intolerance and chronic fatigue. Our results are also consistent with previous studies that showed autonomic vasomotor dysfunction in PASC patients [8,9,10,11,14]. However, compared to previous studies, our study shows a significantly high proportion of POTS among PASC patients who present with orthostatic symptoms. There are varying degrees of proportion of POTS among PASC patients in previous studies [11,14,17]. This is likely in part due to inconsistencies in defining POTS in the clinical setting. For example, there is diurnal variation in heart rate response to tilt table testing, and it is possible that other studies may not have controlled for the time of day of their autonomic testing and may have failed to detect orthostatic tachycardia in some patients. In addition, certain medications, such as beta blockers, can masquerade as excessive orthostatic tachycardia if those medications were not discontinued prior to a tilt table test.

Our study also shows that both POTS and PASC present with severe chronic fatigue, orthostatic intolerance, and post-exercise malaise. Interestingly, palpitations were perceived only in some, but not all patients with POTS and PASC, despite the presence of excessive tachycardia. In all cases of PASC and POTS, palpitations were not the main complaints. Instead, fatigue and post-exercise malaise were reported as the primary symptoms. Overall, there were no specific symptoms that were exclusively present in either PASC or POTS patients. It is also possible that some symptoms only appear at certain stages during the disease process. For example, not all POTS patients complained of GI symptoms at the time of evaluation, but all of them reported having had GI symptoms at some point during their clinical course. PASC is a relatively new clinical syndrome, so it is possible that these patients will eventually experience the full spectrum of POTS symptoms in the future.

It is important to note that both PASC and POTS are clinical syndromes with potentially heterogenous etiologies. In our retrospective study, PASC patients who were suspected of having dysautonomia were recruited for analysis. Therefore, our study population likely represents a certain subset of PASC patients. Even among PASC patients in our study, there appears to be some heterogeneity. For example, two patients with PASC showed evidence of cardiovagal and central baroreflex involvement, suggesting the involvement of the parasympathetic and/or central nervous system in addition to the peripheral sympathetic vasomotor nervous system because the cardiovagal reflex center is located in the brain stem. The rest of the PASC patients and all the POTS patients in our study showed some evidence of peripheral sympathetic vasomotor nerve involvement without signs of central nervous system involvement, given the normal heart rate variability in deep breathing. Our study also shows that about half of both PASC and POTS patients had small fiber neuropathy in their skin biopsies. This is consistent with previous reports [11,14]. It is postulated that the systemic involvement of small fiber neuropathy underlies sympathetic and/or parasympathetic nerve damage, given that the majority of peripheral autonomic nerve fibers are unmyelinated small axon fibers. Although intraepidermal small fiber density does not directly evaluate autonomic nerve fibers, it may be used as a surrogate measure for autonomic nerve innervation in an appropriate clinical context. POTS is also known to be associated with non-length dependent pattern small fiber neuropathy [18]. Therefore, “normal” skin biopsies could have been simply due to sampling bias, as we harvest punch skin biopsy samples from standard sites in lower extremities along length-dependent areas.

None of the PASC patients had similar symptoms of chronic fatigue, post-exercise malaise, or orthostatic intolerance prior to the COVID-19 infection. There are a few possibilities that can explain the post-infectious onset of the symptoms. First, a subset of PASC can be explained by autoimmune inflammation targeted against the autonomic nervous system. A few previous studies have shown increased inflammatory and autoimmune markers in PASC patients [19,20]. In our study, two out of seven patients with PASC who received antibody testing had positive autoantibody markers (anti-TPO antibody and anti-ganglionic acetylcholine receptor (gAChR) antibody). However, the anti-TPO antibody is phenotypically associated with Hashimoto’s thyroiditis, but not autonomic neuropathy. The patient with positive anti-TPO antibodies had normal thyroid function and had not been diagnosed with thyroid disease. It is not clear whether anti-TPO antibody is causally related to the patient’s PASC or not. The anti-gAChR antibody is known to be related to autoimmune autonomic ganglinopathy, but its association with POTS has been questioned, especially at low titers < 0.2 nmol/L [21]. It is possible that a novel autoantibody is present in a subset of PASC patients presenting with dysautonomia. Second, some symptoms of PASC can be explained by permanent or longstanding tissue damage. In the case of PASC with dysautonomia, it is possible that SARS-CoV-2 virus can directly invade and damage sympathetic and/or parasympathetic ganglia or axons. Axonal regeneration from neuronal injury is a slow process and often incomplete. In this case, patients have a monophasic course of neurological symptoms over time [22]. However, all the PASC patients in our study have shown a waxing and waning course of neurological symptoms, making this theory less likely. In addition, to date, there has been no autopsy study that has examined peripheral sympathetic chain ganglia in PASC patients. Lastly, PASC can be due to a persistent viral infection that results in chronic inflammation of the autonomic nervous system. There has been conflicting evidence as to whether the SARS-CoV-2 virus can infect the human nervous system or not. However, it appears that most autopsy studies have not shown clear evidence of the direct invasion of SARS-CoV-2 virus in the central nervous system, and inflammatory changes observed in the brain tissue are likely from systemic inflammation [23,24,25,26,27]. As stated above, there have been a relatively small number of studies that have examined neuroinflammation in peripheral autonomic nerve tissues [28]. Considering the implication of PASC on public health, more studies should be undertaken to examine the underlying neuropathology of PASC in the future.

This study has a number of limitations. This is a retrospective study of consecutive patients who were referred to the Johns Hopkins Autonomic Laboratory over a 12-month period. There is likely a selection bias in this study, as certain patients with PASC were referred to the Autonomic Laboratory only when they presented with autonomic symptoms. There is also a relatively small number of patients in this study. During the most part of 2021 and 2022, our institution maintained very strict COVID-19-related restrictions on outpatient visits, which limited the recruitment for the study. In addition, not all patients underwent autoantibody testing and skin biopsy. A large, prospective study will help shed more light on the clinical and autonomic function characteristics of PASC and its relationship with POTS.

## 5. Conclusions

In this retrospective comparative study, the clinical and autonomic features in patients with PASC and POTS were indistinguishable, suggesting a similar underlying disease pathogenesis. Future large and prospective studies will be valuable in further evaluating this association.

## Figures and Tables

**Figure 1 jcm-12-00073-f001:**
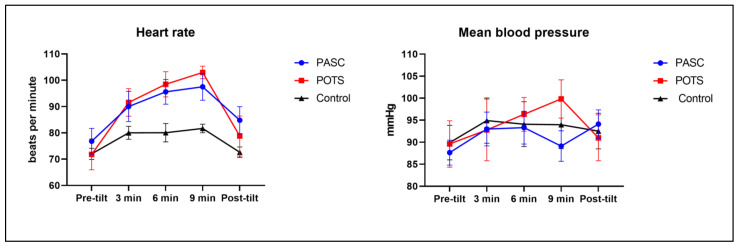
Heart rate and mean blood pressure during 10 min tilt table test. Heart rate and blood pressure were measured at pre-tilt, 3, 6, 9, and post-tilt. Error bars present standard errors of the mean (SEM). Heart rate changes between POTS and control were significantly different (*p* = 0.03) in repeated-measures analysis of variance (ANOVA) analysis. Heart rate changes between PASC and POTS, and PASC and control were not significantly different. Mean blood pressure remained unchanged throughout tilt table test in PASC, POTS, and control group (PASC n = 11; POTS n = 6; control n = 7). PASC stands for post-acute sequelae of SARS-CoV-2 syndrome; POTS stands for postural orthostatic tachycardia syndrome.

**Figure 2 jcm-12-00073-f002:**
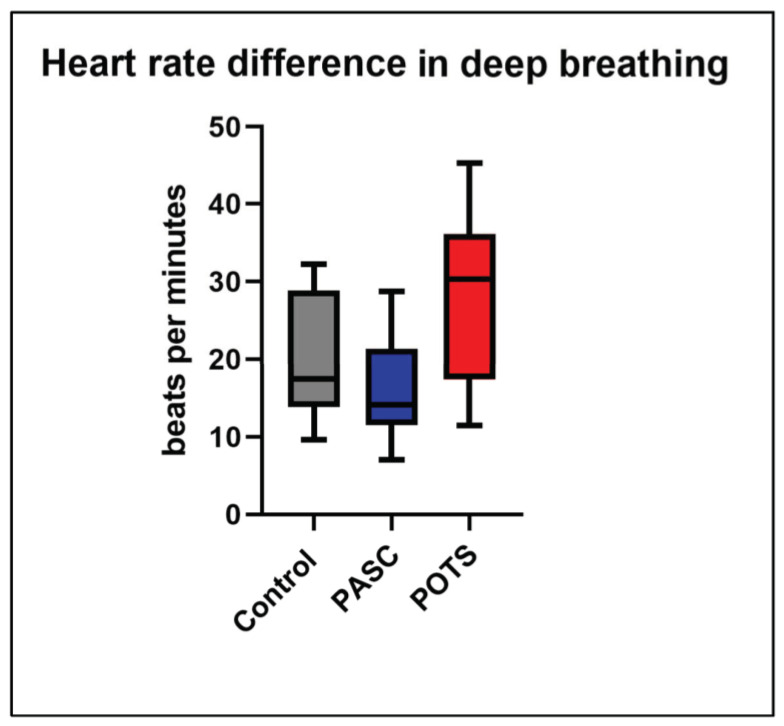
Heart rate variability in deep breathing. The maximal heart rate differences between inspiration and expiration in PASC, POTS, and control patients are presented in box-and-whisker plots (PASC n = 12; POTS n = 6; control n = 7). There was no statistical significance between PASC vs. POTS, PASC vs. control, and POTS vs. control in the *t*-tests. PASC stands for post-acute sequelae of SARS-CoV-2 syndrome; POTS stands for postural orthostatic tachycardia syndrome.

**Table 1 jcm-12-00073-t001:** Demographic features. Standard deviations are presented in the parentheses.

	Mean Age	Sex	Ethnicity	Mean BMI
Post-Acute Sequelae of SARS-CoV-2 Syndrome (PASC) (N = 13)	47.39 (±14.01)	11/13 (84.62%)	White—10/13 (76.92%) Hispanic—1/13 (7.69%) Asian—2/13 (15.38%)	26.93 (±6.79))
Postural Orthostatic Tachycardia Syndrome (POTS) (N = 6)	30.17 (±7.47)	5/6 (83.33%)	White—5/6 (83.33%) Asian—1/6 (16.67%)	26.62 (±7.66))
Control (N = 7)	44 (±16.16)	5/7 (71.43%)	White—5/7 (71.43%) Asian—2/7 (28.57%)	25.69 (±4.81)

PASC: Post-Acute Sequelae of SARS-CoV-2 Syndrome (PASC). POTS: Postural Orthostatic Tachycardia Syndrome.

**Table 2 jcm-12-00073-t002:** Clinical features of PASC and POTS patients following COVID-19 infection. Standard deviations are presented in the parentheses. (N/A: not applicable).

	Post-Acute Sequelae of SARS-CoV-2 Syndrome (PASC)	Postural Orthostatic Tachycardia Syndrome (POTS)	*p* Values for *t*-Test between PASC and POTS	Control
Mean duration of symptoms in months	21.15 (± 5.15)	138.83 (± 123.16)	<0.01	N/A
Severity of COVID-19 infection	Mild—11/13 (84.62%) Moderate—2/13 (15.38%)	N/A	N/A	N/A
Fatigue	13/13 (100%)	6/6 (100%)	N/A	None
Orthostatic intolerance	13/13 (100%)	6/6 (100%)	N/A	None
Palpitations	8/13 (61.54%)	3/6 (50%)	0.64	None
Shortness of breath	9/13 (69.23%)	4/6 (66.67%)	0.90	None
Chest pain	3/13 (23.08%)	0/6 (0%)	0.20	None
Headache	5/13 (38.46%)	3/6 (50%)	0.64	None
Brain fog	12/13 (92.30%)	6/6 (100%)	0.48	None
Diffuse pain	7/13 (53.85%)	2/6 (33.33%)	0.41	None
Gastrointesstinal symptoms	9/13 (69.23%)	3/6 (50%)	0.42	None
Post-exertional malaise	11/13 (92.30%)	6/6 (100%)	0.48	None
Insomnia	10/13 (76.92%)	4/6 (66.67%)	0.64	None

PASC: Post-Acute Sequelae of SARS-CoV-2 Syndrome (PASC). POTS: Postural Orthostatic Tachycardia Syndrome.

## Data Availability

The data that support the findings of this study are available on request from the corresponding author, T.H.C.

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
