# Peer review of "Autonomic Nerve Involvement in Post-Acute Sequelae of SARS-CoV-2 Syndrome (PASC)"

_jcm, 2022, doi:10.3390/jcm12010073_

Round 1

Reviewer 1 Report

This is a retrospective study performed that compares the outcomes of autonomic testing in patients with post-acute sequelae of COVID-19 (PASC) versus those with postural orthostatic tachycardia syndrome (POTS).  Test results from a total of 13 patients with PASC and 6 with POTS were compared, as well as 7 control patients.  The results from the PASC and POTS patients were essentially found to be equivalent.

Although a study like this is necessary to further prove that at least a subset of PASC seems to be equivalent to POTS, there are some concerns about these data and their presentation.  Considering how large and well known this institution’s clinic is, as well as the large number of affected patients with both PASC and with POTS, it is quite surprising (and disappointing) that the total number of patient subjects in this retrospective review was only 19 in the span of one year.  How is it that there were not many more patients in either category who hadn’t undergone complete autonomic testing there?  Certainly, not every patient with PASC had orthostatic intolerance, so that would reduce the number of patients who met inclusion criteria, but a much larger number would be expected.  The authors should better explain why their numbers are so low, despite a year-long inclusion range.

No statistical analysis was performed to demonstrate equivalence, or its lack, in Table 1.  Specifically, the POTS patients appear to be much younger than the PASC patients, but it is not officially demonstrated.  Statistical analysis of the data should be performed.  As well, the data for age and BMI should be displayed as mean +/- SD or median +/- IQR, depending on whether or not they are normally distributed.  Using 95% confidence interval here is inappropriate.  Ethnicity should also include percentages.  The same comments go for Table 2.

Also in Table 2, the authors report on severity of COVID infection, with most patients having mild infection.  However, they do not discuss how they determined the degree of severity in the Materials and Methods section.  This should be addressed.

The patient criteria for inclusion in the study included age 18 years and up, with HR criteria of at least 30 bpm with tilt table testing.  The problem here is that the heart rate threshold for patients age 18 and 19 is at least 40 bpm.  Therefore, any patients under age 20 may not have officially met criteria for the diagnosis of POTS.

For some reason, the authors failed to include articles when enumerating various subjects and objects in sentences.  There were also inappropriate changes in tense and awkward sentence construction.  Once corrections are made, this manuscript needs to be closely re-reviewed by a colleague for sentence construction and word choice prior to resubmission.

Line 37: although autonomic dysfunction may underlie some patients’ symptoms of PASC, there is also published suggestion that mast cell activation syndrome can be the underlying pathophysiology.

Line 74: Diagnostic criteria for POTS in patients age 12-19 include a 40 bpm increase, or more, but the authors included patients down to age 18 years using a 30 bpm increase as threshold.  This needs to be addressed.

Line 76: POTS is not just defined as the heart rate increase.  There must be symptoms of orthostatic intolerance for at least 3 months, and other similar diagnoses are ruled out.

Line 77: where are the data to support the sustaining of a HR increase at least 30% of the 10 minute tilt duration as diagnostic for POTS?

Line 79: the use of the term, “ruled out,” in this sentence is awkward, suggesting that the patients themselves ruled out their diagnosis of autonomic disorders.

Line 82: the use of the phrase, “a full autonomic testing equipment,” is awkward and should be rephrased.

Line 84: should be, “A CNAP monitor…”

Line 85: should be, “…using a CNAP finger sensor.”

Line 86: should be, “…obtained from the finger sensor…”

Line 86 : should be, “…measurements from the antecubital fossa…”  However, unless this was a manual blood pressure measurement, in which the auscultation is performed in the antecubital fossa, the cuff is actually placed on the upper arm.  This needs to be better stated and described.  Does this also mean that the blood pressure in the cuff was being measured intermittently while the finger sensor was also being used?

Line 87: should be, “…an additional blood pressure cuff…”

Line 89: should be, “A chest expansion bellow…”

Line 90: should be either, “A tilt table test…,” or, “Tilt table testing…”

Line 93: should be, “… and adjustable light system to improve patient comfort…”

Line 101: should be, “…with a mouthpiece connected to the WR Medical…”

Line 102: “blow hard” is awkward, and redundant, since the actual pressure is listed—maybe just use the word, “blow,” or, “forcefully exhale.”
Line 105: although most people who perform autonomic testing are familiar with the 4 phases of the Valsalva maneuver, it would be helpful to briefly list (and/or describe) these phases for those readers who do not do this testing.  Also, should be, “The Valsalva ratio…”

Line 107: please spell out, “seconds.”

Line 115: should reference the “established counting rules.”

Line 120: should be either, “…as being an abnormal skin biopsy,” or, “…as being abnormal.”

Line 129: should be, “Box and whisker plots were used…”

Line 131: should be, “Microsoft Excel.”

Line 137: why list the number of PASC patients, but not POTS patients?  Also, it is difficult to believe that only 13 PASC and 6 POTS patients, considering that the authors suggest that there is such a large population of patients with PASC, and that a significant proportion of them have autonomic dysfunction

Line 143: Table 1 lacks any statistical analysis to demonstrate equivalence (or lack of equivalence) between groups in the various demographic features.  Also a closing parenthesis is missing at the end of the 95% confidence intervals for BMI in control patients.

Line 143: Table 1—the authors incorrectly use the term, “gender,” when actually referring to sex.  This should be changed.

Line 143: Table 1 suggests that the POTS patients are significantly younger than the PASC and control groups.

Line 143: Table 1—the table does not state whether the numbers displayed are the mean or median.  These numbers should be displayed as mean +/- SD or median +/- IQR, depending on whether the data are normally distributed, or not.  They should not be displaying 95% confidence intervals.

Line 154: “indigestion” is often taken to be synonymous with gastroesophageal reflux.  Therefore, since acid reflux is listed separately, how do the authors define this term?

Line 155: why the amount of post-exertional malaise in PASC patients mentioned, but not in POTS patients?  Also, the text says that PEM occurred in 11/13 PASC patients, but the table says 12/13.

Line 159: “complained over,” is awkward—should be “complained of.”

Line 160: why did the sentence change to present tense? Should be, “…symptoms that were essentially…”

Line 160: should be, “…noteworthy that all POTS…”

Line 164: why did only 7/13 patients get antibody testing?

Line 169: Table 2—should be a column demonstrating p value differences between the PASC and POTS patients.

Line 169: Table 2—should be palpitations

Line 172: should be, “…exceeded the table manufacturer’s recommendation…”

Line 179: should be, “The remainder of the patients…”

Line 180: should be, “…had an increase in heart rate which did not…”

Line 182: should be, “…or between PASC and control…”

Line 197: “normal best heart rate difference” is awkward.

Line 201: should be, “…normal heart rate differences…”

Line 235: should be, “…shows a significantly high…”

Line 240: active standing masquerading orthostatic tachycardia—this is a very misleading statement, especially when referencing the Plash article.  The sensitivity for diagnosis by 30 point HR threshold was equivalent between the two modalities.  And, tilt table test is not a physiologically normal condition, whereas active stand is.  However, another reason that POTS was likely not diagnosed by other clinics is that, unlike your study (which was done correctly), there is diurnal variation in heart rate response, and the other studies may not have controlled for time of day of their autonomic testing.

Lines 242, 244: palpitations are typically referred to in the plural, not singular.

Line 246: should be, “…in either PASC or…”

Line 305: should be, “There is also a…”

Author Response

This is a retrospective study performed that compares the outcomes of autonomic testing in patients with post-acute sequelae of COVID-19 (PASC) versus those with postural orthostatic tachycardia syndrome (POTS).  Test results from a total of 13 patients with PASC and 6 with POTS were compared, as well as 7 control patients.  The results from the PASC and POTS patients were essentially found to be equivalent.

Although a study like this is necessary to further prove that at least a subset of PASC seems to be equivalent to POTS, there are some concerns about these data and their presentation.  Considering how large and well known this institution’s clinic is, as well as the large number of affected patients with both PASC and with POTS, it is quite surprising (and disappointing) that the total number of patient subjects in this retrospective review was only 19 in the span of one year.  How is it that there were not many more patients in either category who hadn’t undergone complete autonomic testing there?  Certainly, not every patient with PASC had orthostatic intolerance, so that would reduce the number of patients who met inclusion criteria, but a much larger number would be expected.  The authors should better explain why their numbers are so low, despite a year-long inclusion range.

We appreciate the reviewer’s valid comment that only 19 patients were recruited for this study over the span of one year. However, we would like to point out that our institution maintained very strict COVID-related restrictions on in-person outpatient visits for the most part of 2021 and early part of 2022. We had just re-opened the autonomic laboratory in early 2021 and scheduled a very limited number of patients over that time frame, although we continued to see a large number of post-COVID POTS patients via telemedicine. In addition, we only selected patients who had had symptoms more than 4-12 weeks per definition of PASC. Considering that the Omicron surge in the United States started late 2020 and early 2021, we started seeing most PASC patients from omicron variants starting the summer of 2021 under the COVID-related restrictions. Notably, we are not alone in this – many of the recent studies that evaluated autonomic functions of PASC during a similar time period recruited even smaller numbers of patients. We addressed this point in the Discussion of our revised manuscript.

No statistical analysis was performed to demonstrate equivalence, or its lack, in Table 1.  Specifically, the POTS patients appear to be much younger than the PASC patients, but it is not officially demonstrated.  Statistical analysis of the data should be performed.  As well, the data for age and BMI should be displayed as mean +/- SD or median +/- IQR, depending on whether or not they are normally distributed.  Using 95% confidence interval here is inappropriate.  Ethnicity should also include percentages.  The same comments go for Table 2.

We agree with the reviewer’s comment in regards to using mean +/- SD, not 95% CI, in Table 1 and we updated it accordingly in the revised manuscript. We included ethnicity in the percentages as well. We have analyzed t-test for age comparison between POTS and PASC and mentioned it in line 141. We found no statistical differences in other factors, such as ethnicity, gender, or BMI, and updated them in the revised manuscript.

Also in Table 2, the authors report on severity of COVID infection, with most patients having mild infection.  However, they do not discuss how they determined the degree of severity in the Materials and Methods section.  This should be addressed.

We thank the reviewer for their comment. We defined severity of COVID infection as follows: mild – no hospitalization required; moderate – hospitalization required but no ICU stay required; severe – ICU stay required during the course if COVID infection. We clarified the definition in the Materials and Method section in the revised manuscript.

The patient criteria for inclusion in the study included age 18 years and up, with HR criteria of at least 30 bpm with tilt table testing.  The problem here is that the heart rate threshold for patients age 18 and 19 is at least 40 bpm.  Therefore, any patients under age 20 may not have officially met criteria for the diagnosis of POTS.

We have clarified the definition in the Material and Method section. However, the youngest patient was 20 years old at the time of the recruitment, and therefore, the results have not been changed.

For some reason, the authors failed to include articles when enumerating various subjects and objects in sentences.  There were also inappropriate changes in tense and awkward sentence construction.  Once corrections are made, this manuscript needs to be closely re-reviewed by a colleague for sentence construction and word choice prior to resubmission.

Thank you for the advice. We have our manuscript carefully re-reviewed by colleagues for grammatical or spelling errors.

Line 37: although autonomic dysfunction may underlie some patients’ symptoms of PASC, there is also published suggestion that mast cell activation syndrome can be the underlying pathophysiology.

We have included mast cell activation syndrome as a possible underlying pathophysiology.

Line 74: Diagnostic criteria for POTS in patients age 12-19 include a 40 bpm increase, or more, but the authors included patients down to age 18 years using a 30 bpm increase as threshold.  This needs to be addressed.

Please see response above.

Line 76: POTS is not just defined as the heart rate increase.  There must be symptoms of orthostatic intolerance for at least 3 months, and other similar diagnoses are ruled out.

We have updated the definition in our revised manuscript.

Line 77: where are the data to support the sustaining of a HR increase at least 30% of the 10 minute tilt duration as diagnostic for POTS?

In Figure 1., we graphically presented the HR increase that sustained more than 30% of the 10-minute tilt.

Line 79: the use of the term, “ruled out,” in this sentence is awkward, suggesting that the patients themselves ruled out their diagnosis of autonomic disorders.

We included “by referring physicians” to clarify the meaning of the sentence.

Line 82: the use of the phrase, “a full autonomic testing equipment,” is awkward and should be rephrased.

We revised the sentence in the new manuscript.

Line 84: should be, “A CNAP monitor…”

We revised it accordingly.

Line 85: should be, “…using a CNAP finger sensor.”

We revised it accordingly.

Line 86: should be, “…obtained from the finger sensor…”

We revised it accordingly.

Line 86 : should be, “…measurements from the antecubital fossa…”  However, unless this was a manual blood pressure measurement, in which the auscultation is performed in the antecubital fossa, the cuff is actually placed on the upper arm.  This needs to be better stated and described.  Does this also mean that the blood pressure in the cuff was being measured intermittently while the finger sensor was also being used?

We revised it to “upper arm”. Yes, the blood pressure in the cuff was being measured intermittently while the finger sensor was also being used.

Line 87: should be, “…an additional blood pressure cuff…”

We revised it accordingly.

Line 89: should be, “A chest expansion bellow…”

We revised it accordingly.

Line 90: should be either, “A tilt table test…,” or, “Tilt table testing…”

We revised it accordingly.

Line 93: should be, “… and adjustable light system to improve patient comfort…”

We revised it accordingly.

Line 101: should be, “…with a mouthpiece connected to the WR Medical…”

We revised it accordingly.

Line 102: “blow hard” is awkward, and redundant, since the actual pressure is listed—maybe just use the word, “blow,” or, “forcefully exhale.”

We revised it accordingly.

Line 105: although most people who perform autonomic testing are familiar with the 4 phases of the Valsalva maneuver, it would be helpful to briefly list (and/or describe) these phases for those readers who do not do this testing.  Also, should be, “The Valsalva ratio…”

We revised it accordingly.

Line 107: please spell out, “seconds.”

We revised it accordingly.

Line 115: should reference the “established counting rules.”

We added a reference in the revised manuscript.

Line 120: should be either, “…as being an abnormal skin biopsy,” or, “…as being abnormal.”

We revised it accordingly.

Line 129: should be, “Box and whisker plots were used…”

We revised it accordingly.

Line 131: should be, “Microsoft Excel.”

We revised it accordingly.

Line 137: why list the number of PASC patients, but not POTS patients?  Also, it is difficult to believe that only 13 PASC and 6 POTS patients, considering that the authors suggest that there is such a large population of patients with PASC, and that a significant proportion of them have autonomic dysfunction

We revised it accordingly. Please see our response above regarding the number of recruitments. 

Line 143: Table 1 lacks any statistical analysis to demonstrate equivalence (or lack of equivalence) between groups in the various demographic features.  Also a closing parenthesis is missing at the end of the 95% confidence intervals for BMI in control patients.

We updated statistical analysis in the Table 1 in our revised manuscript.

Line 143: Table 1—the authors incorrectly use the term, “gender,” when actually referring to sex.  This should be changed.

We revised it accordingly.

Line 143: Table 1 suggests that the POTS patients are significantly younger than the PASC and control groups.

We now clarified that in the revised manuscript.

Line 143: Table 1—the table does not state whether the numbers displayed are the mean or median.  These numbers should be displayed as mean +/- SD or median +/- IQR, depending on whether the data are normally distributed, or not.  They should not be displaying 95% confidence intervals.

We revised it accordingly.

Line 154: “indigestion” is often taken to be synonymous with gastroesophageal reflux.  Therefore, since acid reflux is listed separately, how do the authors define this term?

Indigestion was defined as the feeling of foods stuck in their stomach and not moving down whereas gastroesophageal reflux was defined as the feeling of heart burn in the stomach.

Line 155: why the amount of post-exertional malaise in PASC patients mentioned, but not in POTS patients?  Also, the text says that PEM occurred in 11/13 PASC patients, but the table says 12/13.

We now mentioned PEM in POTS patients in the revised manuscript. And thank you for the correction – 11/13 is correct.  12/13 was a typo.

Line 159: “complained over,” is awkward—should be “complained of.”

We revised it accordingly.

Line 160: why did the sentence change to present tense? Should be, “…symptoms that were essentially…”

We revised it accordingly.

Line 160: should be, “…noteworthy that all POTS…”

We revised it accordingly.

Line 164: why did only 7/13 patients get antibody testing?

As we mentioned above, most of follow-up visits occurred virtually, and it was very difficult for many patients to come to the lab for blood draw. We hope to be able to obtain more antibody testing in the future.

Line 169: Table 2—should be a column demonstrating p value differences between the PASC and POTS patients.

We revised it accordingly.

Line 169: Table 2—should be palpitations

We revised it accordingly.

Line 172: should be, “…exceeded the table manufacturer’s recommendation…”

We revised it accordingly.

Line 179: should be, “The remainder of the patients…”

We revised it accordingly.

Line 180: should be, “…had an increase in heart rate which did not…”

We revised it accordingly.

Line 182: should be, “…or between PASC and control…”

We revised it accordingly.

Line 197: “normal best heart rate difference” is awkward.

We changed the phrase in the revised manuscript.  

Line 201: should be, “…normal heart rate differences…”

We revised it accordingly.

Line 235: should be, “…shows a significantly high…”

We revised it accordingly.

Line 240: active standing masquerading orthostatic tachycardia—this is a very misleading statement, especially when referencing the Plash article.  The sensitivity for diagnosis by 30 point HR threshold was equivalent between the two modalities.  And, tilt table test is not a physiologically normal condition, whereas active stand is.  However, another reason that POTS was likely not diagnosed by other clinics is that, unlike your study (which was done correctly), there is diurnal variation in heart rate response, and the other studies may not have controlled for time of day of their autonomic testing.

This is an excellent point, and we have included diurnal variation in heart rate response as a potential explanation in the revised manuscript.

Lines 242, 244: palpitations are typically referred to in the plural, not singular.

We revised it accordingly.

Line 246: should be, “…in either PASC or…”

We revised it accordingly.

Line 305: should be, “There is also a…”

We revised it accordingly.

Reviewer 2 Report

The topic is of significant interest. However there is a major problem with the conclusions as currently stated. The manuscript is presented as indicating that this small set of patients are representative of long covid and looks like POTS. The problem is selection bias. Patients were selected based on the presence of orthostatic tachycarida or orthostatic hypotension. This selection approach excludes a very large number of long Covid patients. The manuscript has potential but is in need of significant revision.

They should indicate that patients had mild POTS (note HR on tilt was <120bpm)

How was HR on tilt determined? Was it minute by minute average (the appropriate approach) or reading the HR envelope (see original definition in original paper on POTS).

Demonstration of small fiber denervation. Main tool has been skin biopsy which measures mainly somatic C fibers, when the focus is on autonomic C fibers. Was QSART done to evaluate autonomic C fibers?

Discussion on autonomic physiology is long and unfortunately rather sloppy:

  1. Vasomotor impairment. How can you make this conclusion when you did not test vasomotor reflexes?
  2. Central baroreflex impairment. You make this conclusion when baroreflexes were not evaluates and how did you identify site of lesion as central

Discussion on autoimmune inflammatory markers is weak.

  1. Low titers are not meaningful and often found in other conditions. If they make this point they need to define what the threshold of abnormality is. 
  2. Inflammatory markers should be discussed in a more circumspect way. Minor nonspecific abnormalities provide noise rather than clarity.

Discussion of prevalence is inappropriate. They did not undertake a population based study.

Range and severity of autonomic symptoms: They discuss other symptoms in POTS and long Covid. However lack of more quantitative approach such as COMPASS-31 weakens the comparison.

Author Response

The topic is of significant interest. However there is a major problem with the conclusions as currently stated. The manuscript is presented as indicating that this small set of patients are representative of long covid and looks like POTS. The problem is selection bias. Patients were selected based on the presence of orthostatic tachycarida or orthostatic hypotension. This selection approach excludes a very large number of long Covid patients. The manuscript has potential but is in need of significant revision.

We agree that the results of our study may have been influenced by selection bias due to the retrospective nature of the study. We have already mentioned this in our Discussion. It is not our intention to suggest that the small set of our patients analyzed here represent a very large number of long COVID patients. In the revised manuscript, we have clarified this issue. However, we believe that our study results important because 1) many surveys have shown that significant numbers of long COVID patients present with orthostatic intolerance and/or autonomic symptoms and 2) only very few studies have examined autonomic functions in long COVID patients, despite increasing suspicion of autonomic dysfunction in long COVID patients. While we recruited only 13 patients of PASC in our study, we believe that ours is one of the largest studies that examined autonomic functions in PASC patients to date. To our knowledge, most studies of long COVID patients are focused on psychological aspects, and only few studies have attempted to examine underlying pathophysiology in human subjects. 

They should indicate that patients had mild POTS (note HR on tilt was <120bpm)

We respectfully disagree to use the term “mild POTS” based on the number of HR increase because there is no formal way of defining mild/moderate/severe POTS. In fact, many POTS patients have symptoms severe enough to make them disabled while their HR increases only moderately.

How was HR on tilt determined? Was it minute by minute average (the appropriate approach) or reading the HR envelope (see original definition in original paper on POTS).

As described in the Materials and Method section, we used continuous 3-lead ECG monitoring in addition to CNAP monitoring. However, we only showed 3, 6, 9-minute marks in Figure 1 so that readers can see the graph easily.

Demonstration of small fiber denervation. Main tool has been skin biopsy which measures mainly somatic C fibers, when the focus is on autonomic C fibers. Was QSART done to evaluate autonomic C fibers?

Unfortunately, QSART is not being done in our institution due to some regulatory reasons. However, our cutaneous nerve biopsy lab evaluates sudomotor innervation, which is autonomic C fibers. We have excluded sudomotor innervation in our analysis because there is no normative data for comparison. We have used intraepidermal small fiber density as a surrogate measure of autonomic C fiber, although we acknowledge that these nerve fibers have different origins. However, other than QSART or skin biopsy, there are very few clinical tools that can evaluate autonomic C fibers. We have elaborated these points in our revised manuscript. 

Discussion on autonomic physiology is long and unfortunately rather sloppy:

  1. Vasomotor impairment. How can you make this conclusion when you did not test vasomotor reflexes?

We did not draw any conclusions about vasomotor impairment from our data. We only mentioned that our data suggest vasomotor impairment based on hemodynamic changes, skin biopsy results, and symptoms. We are not clear what “vasomotor reflex test” that the reviewer mentioned, but if that means cutaneous vasomotor reflex test, we do not think cutaneous vasomotor reflex test will be able to prove systemic vasomotor impairments underlying POTS (if POTS is caused by vasomotor impairment). Clinical modalities to evaluate for vasomotor function are very limited, and we mentioned this in our revised manuscript.

  1. Central baroreflex impairment. You make this conclusion when baroreflexes were not evaluates and how did you identify site of lesion as central

While we did not draw a conclusion that central baroreflex is intact, the normal heart rate variability during deep breathing suggests that cardiovagal reflex is intact at the brain stem. In addition, there was no upper motor neuron signs to suggests central lesions in both POTS or PASC patients. We have included this in our revised manuscript.

Discussion on autoimmune inflammatory markers is weak.

  1. Low titers are not meaningful and often found in other conditions. If they make this point they need to define what the threshold of abnormality is. 

We have already mentioned the issues with low-titer ganglionic AChR antibody in our Discussion. We mentioned the threshold of abnormality in our revised manuscript.

  1. Inflammatory markers should be discussed in a more circumspect way. Minor nonspecific abnormalities provide noise rather than clarity.

We have already mentioned the nonspecific nature of inflammatory markers in the Discussion section. We will be more than happy to elaborate on this if we have more detailed suggestions.

Discussion of prevalence is inappropriate. They did not undertake a population based study.

We appreciate this comment. We revised it accordingly.

Range and severity of autonomic symptoms: They discuss other symptoms in POTS and long Covid. However lack of more quantitative approach such as COMPASS-31 weakens the comparison.

We agree that our study is limited due to the retrospective nature. We mentioned in our revised manuscript and hope to be able to use quantitative approaches in the future studies.

Round 2

Reviewer 1 Report

The authors have done an excellent job revising this manuscript, correcting the large majority of issues raised by the initial review.  It is much more easily readable.  The errors have been fixed.  The issues with displaying the statistical analysis have been fixed, as well. 

However, there is one minor issue that remains: although the authors updated and clarified the diagnostic criteria for POTS, they still left out the fact that symptoms need to be present for at least 3 months.  Once this is updated, this should be acceptable for publication.

Author Response

Dear Reviewers:

Once again, we deeply appreciate the reviewers for carefully reviewing the manuscript out of your busy schedule. Please see our response below.

Reviewer 1:

The authors have done an excellent job revising this manuscript, correcting the large majority of issues raised by the initial review.  It is much more easily readable.  The errors have been fixed.  The issues with displaying the statistical analysis have been fixed, as well.

However, there is one minor issue that remains: although the authors updated and clarified the diagnostic criteria for POTS, they still left out the fact that symptoms need to be present for at least 3 months.  Once this is updated, this should be acceptable for publication.

  • Thank you very much for pointing out this important issue. We updated the diagnostic criteria for POTS accordingly in the revised manuscript.

Reviewer 2 Report

The main message is that PASC with autonomic symptoms are indistinguishable from POTS. Yet evidence is unconvincing. You need to indicate, in the year of observation, how many patients presented with PASC, how many of those had autonomic symptoms and how many had orthostatic intolerance.  Surely you saw more than 13 patients with PASC.

There continues to be very sloppy scientific writing. Please do not confuse how common something is with prevalence, which specifies a population based study.

The discussion on why difference studies vary in how common POTS is in PASC is rather specious and unconvincing.

Overall, the writing style is rather pompous. A very small study based on 8 patients with 13 with PASC that is retrospective does not deserve this lengthy report. It could easily be accommodated in a manuscript half the size.

Author Response

Dear Reviewers:

Once again, we deeply appreciate the reviewers for carefully reviewing the manuscript out of your busy schedule. Please see our response below.

Reviewer 2:

The main message is that PASC with autonomic symptoms are indistinguishable from POTS. Yet evidence is unconvincing. You need to indicate, in the year of observation, how many patients presented with PASC, how many of those had autonomic symptoms and how many had orthostatic intolerance.  Surely you saw more than 13 patients with PASC.

  • We fully acknowledge that this is a very important question. Unfortunately, we currently do not have the exact statistics on the entire number of patients presented with PASC and how many of those patients have autonomic symptoms and/or orthostatic intolerance. We have seen a large number of PASC patients in our institute but to obtain such data, we needed to have an IRB approval and funding. We recently obtained the IRB approval to collect various data related to autonomic functions from PASC and POTS patients. We plan to prospectively collect the data once our study is funded and hope that the current publication can help with the grant application in the future.

There continues to be very sloppy scientific writing. Please do not confuse how common something is with prevalence, which specifies a population based study.

  • We appreciate this comment. To avoid the confusion, we replaced the word “prevalence” with “proportion” in the revised manuscript.

The discussion on why difference studies vary in how common POTS is in PASC is rather specious and unconvincing.

  • The explanation in the Discussion was suggested by the reviewer 1, which we agreed, and we understand that there are many other reasons that can explain the difference in how common POTS is in PASC, such as medications. We included possible effects of medications that may explain the difference in the revised manuscript.

Overall, the writing style is rather pompous. A very small study based on 8 patients with 13 with PASC that is retrospective does not deserve this lengthy report. It could easily be accommodated in a manuscript half the size.

  • We fully sympathize with the reviewer’s opinion. However, we respectfully disagree with the assessment. PASC is a very new condition that started less than 2-3 years ago. At the same time, it takes typically longer than 3-5 years of preparation, planning, and securing a grant to complete a high-quality clinical study. There has been not enough time for any investigators to complete such high-quality studies on PASC. Despite this, there is an urgent and desperate need to share knowledge on PASC among medical professionals because PASC is a debilitating condition that has swiftly affected millions of people worldwide. Given this, many journals have published small studies related to PASC based on potential impact, but not necessarily based on the size of the studies. In fact, a recent study by Novak et al., which was published in a major neurology journal, is a retrospective study that only recruited 9 PASC patients but has been extensively cited since the publication. We believe that our study has provided important messages to the medical professionals who take care of patients with PASC and is suitable to the current journal’s special issue, "Long COVID: Current Approaches and Clinical Challenges in Treatment and Rehabilitation".
